# Prevalence and Molecular Characterization of *Echinococcus granulosus* Sensu Lato Eggs among Stray Dogs in Sulaimani Province—Kurdistan, Iraq

**DOI:** 10.3390/vetsci9040151

**Published:** 2022-03-22

**Authors:** Hazhar M. Aziz, Abdullah A. Hama, Mariwan A. Hama Salih, Allah Ditta

**Affiliations:** 1Department of Medical Laboratory Technology, Kalar Technical College and Research Center, Sulaimani Polytechnic University, Sulaymaniyah 46001, Kurdistan, Iraq; 2Department of Medical Laboratory, College of Health and Medical Technology and Research Center, Sulaimani Polytechnic University, Sulaymaniyah 46001, Kurdistan, Iraq; mariwan.abdulla@spu.edu.iq; 3Medical Laboratory Science, College of Health Science, University of Human Development, Sulaymaniyah 46001, Kurdistan, Iraq; 4Department of Environmental Sciences, Shaheed Benazir Bhutto University Sheringal, Dir 18000, Pakistan; allah.ditta@sbbu.edu.pk; 5School of Biological Sciences, The University of Western Australia, 35 Stirling Highway, Perth, WA 6009, Australia

**Keywords:** *Echinococcus granulosus*, molecular identification, PCR, stray dogs

## Abstract

The main goal of this study was to estimate the prevalence of *Echinococcus granulosus* among stray dogs, as well as its potential impact on the environmental contamination in the Kurdistan-Iraq using microscopic examination and the Copro-PCR method. The presence of taeniid eggs was recorded in 400 dog faeces collected from the four different regions in the Sulaimani Governorate. The parasite eggs were recovered from fresh and aged faecal samples of the dogs using two isolation techniques, a flotation method (Sheather’s solution, modified; specific gravity: d = 1.27) and a sedimentation method (formal-ether) in which the sediments from dog faeces were collected. Both methods were used for Copro-PCR to detect the presence of *Echinococcus* species egg through DNA using common primers designed to amplify a partial gene of cytochrome oxidase subunit 1 (COX_1_). The results of the microscopic examination showed a higher prevalence rate, i.e., 97 (24.25%) of *E. granulosus* among stray dogs generally in Sulaimani Governorate. The prevalence of *E. granulosus* among stray dogs according to the district area was 40, 24, 23, and 20.8% in Rzgari, Kalar, Sulaimani, and Halabja, respectively. The positive samples (*n* = 50) were selected for molecular confirmation, the DNA was extracted from the sediment of the positive samples and 40 (80%) samples were successfully amplified by polymerase chain reaction. The sequences show that all samples belong to the *Echinococcus granulosus* sensu lato (G_1_–G_3_), with slight genetic variation. It was concluded that the sediment of dog faeces can be used for DNA extraction, which is a new method that increases the sensitivity of the test, and the amount of DNA yield would be higher than the routine method, which directly uses faeces of the dogs. In addition, the molecular diagnosis was more sensitive than the microscope examination for the presence of *E. granulosus* eggs. The prevalence of *E. granulosus* in both the final hosts and the intermediate hosts must be regularly monitored.

## 1. Introduction

A dog or a cat is often seen in close quarters with humans. The number of stray dogs, in particular, has increased in various countries around the world. In urban areas, where the dog owners often neglect to clean up dogs’ faeces, this might represent a danger to human health and hygiene. Various kinds of pathogens potentially harmful to humans may be found in dog faeces [1]. *Echinococcus granulosus* is a parasitic pathogen that belongs to the metacestode of the Taeniidae family. Although each genotype may infect many intermediate hosts, these are categorized into various genotypes (G_1_–G_10_), depending on the intermediate host [2]. *Echinococcus granulosus* (s.l.) includes (G_1_–G_3_) genotypes, the G_4_ is *E. equinus*, G_5_ is *E. ortleppi*, and G_6_-G_10_ are referred to as *E. canadensis* [3]. The strains of *E. granulosus* have adapted to different intermediate hosts such as sheep, pigs, cattle, horses, camels, goats, and cervid at least; seven of these strains are infective to humans [4]. The domestic cat (*Felis catus*) is the most abundant carnivore in urban and peri-urban areas with high human density [5]. The cats may be accidentally infected with echinococcosis. A new cystic echinococcosis (CE) case in a cat was reported in Uruguay [6] and recently in Turkey, a similar case of cat CE was reported [7]. The molecular analysis confirmed the cyst has belonged to *Echinococcus granulosus* (G_1_).

Infection with cestodes of the species *Echinococcus* in the metacestode stage causes human echinococcosis, which is a significant public health problem in many parts of the world [4]. The dog is the final host of *Echinococcus granulosus.* The infected dogs shed a huge number of eggs of *Echinococcus granulosus* daily and are the main source of soil, food, and vegetable contamination. Because of the importance of stray dogs in CE transmission, epidemiological and genetic research on these hosts is essential in each endemic location to ensure that control programs are implemented effectively [8]. The human is the dead-end of the disease, as the human-to-dog transmission of the parasite is impossible [9]. Ingesting food or water contaminated with faeces containing *E. granulosus* eggs transmitted from infected animals, or handling a pet or infected dogs, causes humans to be infected [10]. Livestock CE is widespread in many regions of Asia including Middle East countries [11], Central Asia [4], and Iran [12]. Domestic and wild carnivores harbour the mature tapeworm in their small intestine. Intermediate hosts, usually herbivores are infected by ingesting the parasite eggs, which spread in the stool of the definitive host [13].

Upon expulsion, proglottids, which contain eggs of *Echinococcus*, are exposed to the external environment [14]. Stray dogs, as the definitive host of adult *E. granulosus*, play the most important role in the spread of infection, by contaminating the environment with their eggs [8]. Sheep, cattle, buffaloes, camels, goats, donkeys, and pigs act as suitable intermediate hosts [15]. The eggs of *E. granulosus* survive for more than 200 days at 7 °C under humid conditions while the eggs survive for about 50 days at 21 °C under low humidity. In contrast, the eggs survive for only a few hours under hot, dry conditions (40 °C). The temperature and humidity influence egg infectivity but do not control the parasite population [4]. The eggs of *Echinococcus* are very sensitive to desiccation [3]. *Echinococcus granulosus* can survive for a long time under cold environmental conditions however, its growth and development are reduced under these conditions [16].

The use of fresh vegetables has increased the number of reported instances of foodborne infections in recent years. Raw vegetable consumption has a significant epidemiological role in the spread of foodborne pathogenic infections [17]. In developing countries, intestinal parasites are common, likely because of low sanitation and personal hygiene [18]. In some studies, performed around the world, fresh vegetables have been linked to the transmission of helminth eggs and larvae [19,20].

Echinococcosis is a growing concern, possibly due to the worldwide distribution of *E. granulosus*, the difficulty of diagnosis and control, the high cost and complexity of treatment (cyst removal may require surgical intervention). In some cases, finding suitable molecular genetic studies for strain identification of *E. granulosus* obtained primarily through mitochondrial DNA (mtDNA) has an important role in strategic control plans [21]. The infected dog is the main source of environmental contamination with dog tapeworm eggs.

The coproantigen ELISA test has become the best laboratory-based test for mass screening of infection in dogs because it can identify infection with a high level of certainty during the prepatent period [22]. The screening of stray dogs for *Echinococcus* infection is essential to reduce human and animal echinococcosis, which is a silent public health concern [21]. The present study is an attempt to evaluate the level of contamination of dog faeces with the eggs of the dog tapeworm, a neglected disease in Kurdistan. According to our knowledge, this is the first study using a large sample size to determine the prevalence of echinococcosis among stray dogs in Sulaimani province.

## 2. Materials and Methods

### 2.1. Study Area

The present study was conducted in the Kurdistan Region (Sulaimani Centre, Halabja, Kalar, and Rzgary districts) of Iraq located between 35°4′ and 36°30′ latitude and 44°50′ and 46°16′ longitude. The altitude of the Sulaimani governorate is about 400 m above sea level, and 2500 m in the northern and northeastern of the province. The climate of the Kurdistan region is hot summers (the highest temperature is about 46 °C) with cold winters (the minimum temperature is about −5 °C).

### 2.2. Collection of Dog Faeces Samples

Four hundred dog faecal samples were taken from the Sulaimani Centre (*n* = 130), Halabja (*n* = 112), Kalar (*n* = 120), and Rzgary (*n* = 35) from September 2019 to November 2021. The sampling areas were all rural locations where livestock is grown, and stray and semi-feral dogs have been recorded. The survey was not based on a certain number of dogs, but rather on the availability of dog faeces from different areas to find the rate and prevalence of *E. granulosus* among stray dogs. For samples collection, 20 sites were visited in each area during the winter, spring, summer, and autumn seasons. The collected samples were properly labelled and stored in the laboratory at −80 °C for seven days to inactivate the parasite egg’s infective phases until the faeces were analysed [6] and then kept at a temperature of 20 °C.

### 2.3. Isolation of Eggs

The parasite eggs were recovered from faecal samples by direct wet mount microscopic examination and sedimentation technique (formal-ether), followed by centrifugation at 1200× *g* [23]. The taeniid eggs were subsequently identified morphologically [24]. The eggs were collected from the coverslip, using a 0.9% NaCl solution, and stored at −20 °C. Nevertheless, the taeniid eggs are morphologically indistinguishable and the egg identification for *E. granulosus* requires the use of the PCR technique [25].

### 2.4. DNA Extraction

In total, 50 samples, where eggs were isolated as described above, had DNA extracted using a stool DNA extraction kit (Geneaid, Taiwan) following the manufacturer’s instructions with some modification. About 200 mg of the dog faeces were filtered through a medical gauze (Size: 10.2–10.4 cm), to remove the large particles and the eluent was washed three times with normal saline by centrifugation for 5 min at 3000 rpm each time, the supernatants were discarded. Finally, formal-ether sedimentation techniques were followed [26], and about 5 mg of the sediment was processed for DNA extraction with the faecal DNA extraction kit. This method was developed by the authors to increase DNA yield.

### 2.5. PCR Amplification and Sequencing

The PCR was performed to detect the target DNA of *E. granulosus* and another taeniid, the previously described [27], primers i.e., Eg-Cox1F (5′-TTTTTTGGGCATCCTGAGGTTTAT-3′) and Eg-Cox1R (5′-TAAAGAAAGAACATAATGAAAATG-3′) for *E. granulosus* were used. The expected PCR product band was 444 bp. PCR was carried out in a final volume of 25 μL,12 mL SuperAdd Taq Master Mix (Promega, Madison, WI, USA), 1 μL of each primer (10 pmol uL^−1^), 7 μL H_2_O, 4 μL of template DNA (100–200) ng. The PCR conditions were as follows; 5 min. at 94 °C (initial denaturation), 35 cycles of 30 s at 94 °C, 45 s at 50 °C, 40 s at 72 °C, and finally 10 min. at 72 °C (final extension). The PCR products were analysed by electrophoresis on 1% agarose gel stained with a safe dye stain (SinaClon, Iran). A 100 bp DNA ladder as a molecular size marker was run together with samples to determine the fragment lengths, the PCR products were sent for DNA sequencing (Macrogen, Korea) and the result was assembled and aligned with the previous record in GeneBank using BioEdit (7.2) software program. All sequences were submitted to GeneBank.

### 2.6. Statistical Analysis

The statistical analysis of the collected data was performed using the statistical package GraphPad Prism (version 7.02). The chi-square test was used to find out the differences between categorical variables from a random sample to judge the goodness of fit between expected and observed results, and the significance level applied was 5%.

## 3. Results and Discussion

Coprology studies using microscopes of faecal samples from a dog were used to determine the presence of Taeniid eggs by indirect concentration techniques i.e., sedimentation and floatation methods (Figure 1).

The data revealed that out of the 400 total samples examined, 97 (24.25%) were positive for taeniid parasites. *Echinococcus*
*granulosus* was more frequently detected in stray dogs in the Sulaimani city centre and Halabja city (Suburban area) which were 7.5 and 7.0%, respectively, than in Kalar district and Rzgari (suburban area) which were 6.25 and 3.5%, respectively in Kurdistan (Sulaimani governorate) (Table 1).

The prevalence of the *Echinococcus* eggs among stray dogs was calculated according to the district. The highest prevalence was found in the Rzgari area 14/35 (40%) while the highest rate was recorded in Kalar district 28/115 (24.3%) (Figure 2).

Our findings showed a high prevalence rate of echinococcosis among stray dogs in the Sulaimani governorate. The prevalence in stock animals in the same area was 11.5% [28], which explained the high prevalence of echinococcosis among slaughtered animals. The high prevalence of echinococcosis among stray dogs may be due to a lack of proper management of the infected organs. The high prevalence rate of dog tapeworm infection among dogs may also be because infected dogs are the main source of environmental contamination with worm eggs, and the animal may get the disease after ingestion of the egg with contaminated grass and vegetables. In addition, Bajalan et al. [14] recorded a high prevalence rate (20%) among stray dogs. The direct access of the infected organs of animals for stray dogs was suggested to be the main cause of the high prevalence rate among stray dogs. Similar results of high prevalence rate (19.1%) were reported by Kohansal et al. [29] in Iran. The prevalence rate of echinococcosis among stray dogs varied from high prevalence as in the Kalar city from Iraq [8] recorded 78%, and also Khamesipour et al. [30] reported the high prevalence rate of echinococcosis among stray dogs from 1985 to 2019 in Iran. The average rate of infection was 11.28% while some studies recorded a lower prevalence rate in comparison to our findings; Kawasmeh et al. [31] reported 15% of infection among stray dogs in Saudi Arabia. This variation might be due to environmental factors, population cultures, awareness, and knowledge about the hydatid disease. In poor communities, the awareness and knowledge about dog tapeworms are at a low level. Still, the shepherd has the untreated dog (may be infected with dog tapeworm) with their sheep, the animals are slaughtered outside the slaughter house and without official veterinarian inspection and sold illegally. All these factors might be responsible for the increased prevalence rate of echinococcosis among stray dogs.

The high prevalence rate of *Echinococcus* species among stray dogs in an urban area may also be because stray dogs that are free, living in the streets, can easily reach the offal, and infected organs of the slaughtered animals. In addition, there is a low level of awareness and education of people about this disease and the management of the slaughtered animal offal and wastes. The current study is the first survey using dog faeces examination to determine the prevalence rate of echinococcosis among stray dogs in Sulaimani province.

Fifty positive samples (dog faeces) were subjected to molecular diagnosis technique (polymerase chain reaction) which has high sensitivity and specificity for *E. granulosus* eggs detection. Due to the high morphological similarity of the taeniid eggs, PCR is required for differentiation between *E. granulosus* and other taenia eggs. The PCR results reveal 40/50 (80%) positive (Figure 3), which means that only 80% of the egg identified by microscopic examination belong to *E. granulosus* (Table 2). The results of the DNA sequencing showed that the *Echinococcus granulosus* sensu lato was the common strain responsible for dog infection. The reported strains were deposited in GenBank under the following accession numbers: OM200173, OM200174, OM200175, OM200176, and OM200177. All the sequences with a slight variation belonged to the *E. granulosus* sensu lato and many studies in the same region on the human and slaughtered animal support our findings. The studies have found the common strains (G_1_ and G_3_) which are responsible for human and animal echinococcosis [21,32,33].

The previous study indicated the PCR and molecular diagnosis method is valuable in the molecular genotypic characterization of *Echinococcus* species [32]. The sheep strain (G_1_) is the most frequent in Kurdistan, Iraq, and it may infect a wide range of intermediate hosts, including sheep, goats, cattle, and humans [33]. The PCR findings revealed that 40 (80%) from 50 samples were infected with *E. granulosus*. Many molecular investigations in Iran [34], China [35], Greece [36], India [37], and Iraq [32] back up the present research findings. According to them, the cytochrome oxidase subunit 1 (COX_1_) has the capacity and effectiveness to distinguish between strains and genetic variants. Our findings emphasize that molecular markers are more sensitive for accurate diagnosis and differentiation between Taenia species.

## 4. Conclusions

The results showed that taeniid infections were widespread among stray dogs in Sulaimani Province, in the Kurdistan region of Iraq. Dog faeces in the environment, especially those of stray dogs, are a major source of infection that may be transmitted to ruminants as intermediate hosts and humans as opportunistic hosts. Infection rates of this parasite vary widely throughout Iraq for a variety of reasons, including the presence or absence of definitive and intermediate hosts. As a result, more research is needed to confirm the accurate incidence of these parasites in all hosts in different locations around the study regions. The most common strains in the Kurdistan Region of Iraq are *E. granulosus* sensu lato (G_1_–G_3_).

## Figures and Tables

**Figure 1 vetsci-09-00151-f001:**
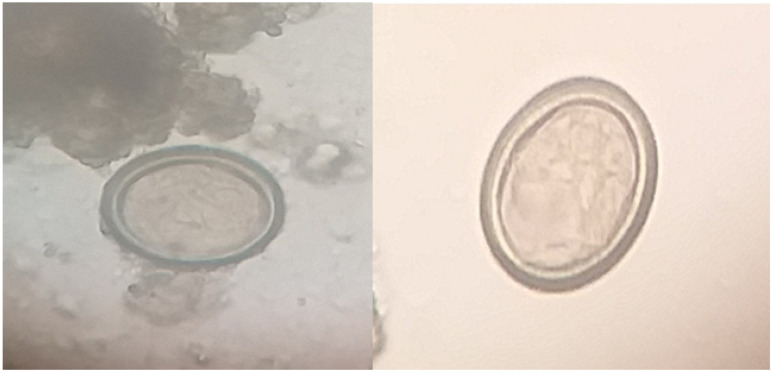
Taeniid eggs (*Echinococcus granulosus*) detected in dog faecal samples (sedimentation technique) 100×.

**Figure 2 vetsci-09-00151-f002:**
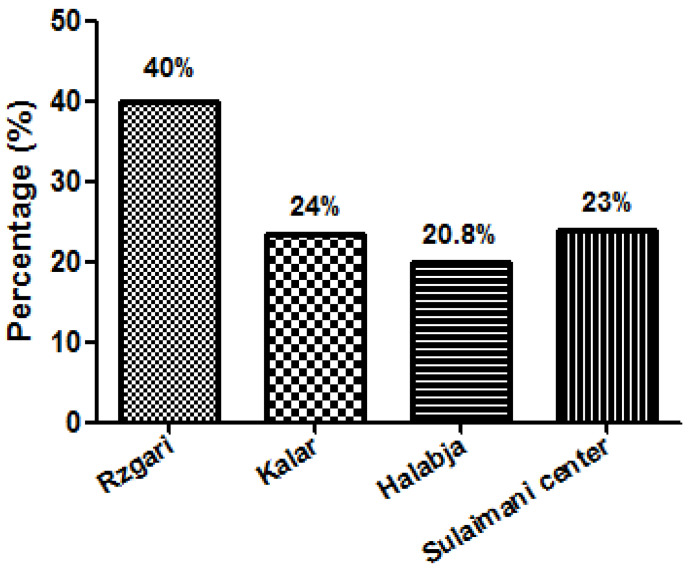
The prevalence rate of *Echinococcus granulosus* egg among stray dogs in different areas of Sulaimani Governorate, Kurdistan—Iraq.

**Figure 3 vetsci-09-00151-f003:**
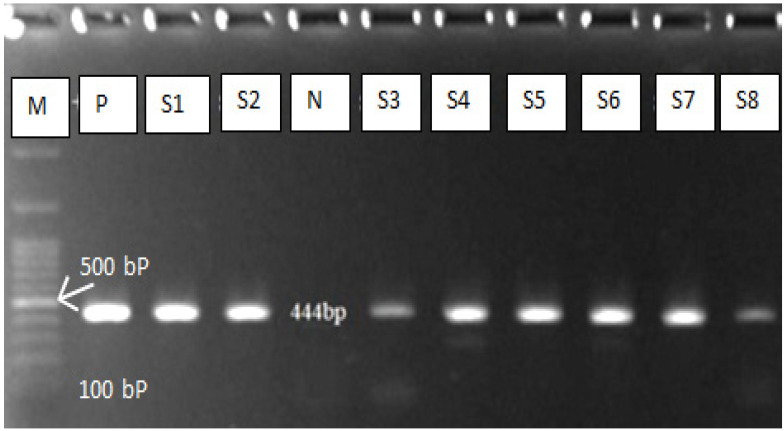
Gel electrophoresis of PCR product of *E. granulosus*. M = DNA Ladder, N = negative control, P = positive control, S = Sample.

**Table 1 vetsci-09-00151-t001:** The prevalence rate of *Echinococcus granulosus* among stray dogs in Sulaimani Governorate using microscopic examination of dog faeces.

Area	Total Samples	Negative Samples	Positive Samples	Percentage of Positive from Total Samples	Statistical Analysis Chi-Square Test
Sulaimani City	130	100	30	7.50	χ^2^ = 5.58 *p* = 0.1335
Halabja	115	87	28	7.00	
Kalar	120	95	25	6.25	
Rzgari	35	21	14	3.50	
Total	400	303	97	24.25	

**Table 2 vetsci-09-00151-t002:** The molecular identification of *Echinococcus granulosus* egg among stray dogs in Sulaimani Governorate.

Study Area	DNA Extraction from Dog Faecal Samples	Positive by PCR
Sulaimani	15	14 (93.3%)
Halabja	12	10 (83.3%)
Kalar	13	11 (84.6%)
Rzgari	10	5 (50%)
Total	50	40 (80%)

## Data Availability

The data presented in this study are available on request from the corresponding author.

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
