# Peer review of "Prevalence and Molecular Characterization of Echinococcus granulosus Sensu Lato Eggs among Stray Dogs in Sulaimani Province—Kurdistan, Iraq"

_vetsci, 2022, doi:10.3390/vetsci9040151_

Round 1
Reviewer 1 Report
Dear authors,
I have found your work timely and appropriate for Vet Sci. Echinococcus granulosus is a big issue as well as other pathogens hosted and potential transmitted by street cats and dogs. I encourage the authors to include a paragraph about this animal health challenge about strat and free-roaming cats and dogs in the introduction section. In the attached file I have included some comments.
I hope that my comments will be useful for your investigation

Author Response
Response to Reviewer 1 comments
Comment: Dear authors, I have found your work timely and appropriate for Vet Sci. Echinococcus granulosus is a big issue as well as other pathogens hosted and potential transmitted by street cats and dogs. I encourage the authors to include a paragraph about this animal health challenge about strat and free-roaming cats and dogs in the introduction section.
Response: Thanks for your comments and recommendation. The paragraph about cat and Echinococcosis was added (Please see lines 48-59)
Comment: In the attached file I have included some comments. I hope that my comments will be useful for your investigation
Here is important to underline the need of conducting health surveys in stray pets. See an example in cats
Candela MG, Fanelli A, Carvalho J, Serrano E, Domenech G, Alonso F, Martínez-Carrasco C. Urban landscape and infection risk in free-roaming cats. Zoonoses Public Health. 2022 Feb 7. doi: 10.1111/zph.12919. Epub ahead of print. PMID: 35129882.
Response: Thanks for your comment. We have consulted the suggested article and cited it (Please see line 53)
Thanks for your valuable comments

Reviewer 2 Report
Introduction
I would start with echinococcus and lead into the hosts and open defecation of dogs particularly as a risk factor for human infection with echinococcosis – and other helminths.
What do you mean by semisenescent?
Page two: these sentences on egg survivability should be rearranged to flow better and make more sense. For instance the first sentence looking at 7C and 21C, is this dry or humid? It then switches in the third sentence to talk about eggs surviving for weeks on months in humid environments – presumably with a temp above the 21C from the first sentence.
Page 2 paragraph 2, is there a reference for increase in food borne infections linked to fresh vegetables? Reference 7 is a single Russian paper, if you are stating around the world you need to have references that reflect and give that information.
Page 2, third paragraph, Introduce the idea of genotypes before stating that each genotype may infect many hosts.
The disease echinococcosis does not need to be in italics
“Because of the importance of stray dogs in CE transmission” – you haven’t indicated that dogs are important in transmission yet. This is where I would introduce echinococcus at the beginning then introduce dogs and how they are important in transmission.
“Human is the dead-end of the disease since the human-to-human or dog-to-human transmission of the illness is impossible” So dogs are not important to transmission after all? I think you mean human to dog (and it is possible, just unlikely as it would require a dog to eat a human). Also the reference would not seem to support this claim.
“Livestock CE is widespread in many regions of Asia including Middle East countries (Iran, Iraq, Pakistan, and Saudi Arabia), Central Asia (Kazakhstan, Kyrgyzstan, Tajikistan, Turkmenistan, and Uzbekistan), China, India, and Japan [13,14].” Refs 13 and 14 are for Nigeria and Iran. I assume they mention that it’s in other countries, but would be best to read some articles from these papers that are listed and then ref
“In some cases, the finding suitable molecular genetic studies for strain identification of E. granulosus obtained primarily through the use of mitochondrial DNA (mtDNA),) which has an important role and is a new clue for the strategical control plan” reference 15 does not use molecular tools. So why is it referenced?
“The infected dog is the main source of environmental contamination with dog tapeworm egg” full stop here.
Reference 16 is not a specific reference to the ELISA. I also read the article and ELISA was not performed in that study, so why is this reference used?
This introduction is really confused and not at all focused. Consider the story you want to tell. Based on the title it should be an introduction to echinococcus in Iraq and how stray dogs may be important. The referencing throughout is quite poor.
Methods
What year was the study performed
Collection of dog faeces
How many from each location?
“The survey was not based on a certain number of dogs, but rather on the availability of Echinococcus granulosus eggs in the region”. Explain this further. So the prev of echinococcus was already known? If you are cherry picking locations based on having known high prev are you really finding the prevalence?
How many sites?
When you say dog feces samples, were these all environmental?
Once you start talking about recovering eggs you need to start at minimum a new paragraph, better, a new subheading for egg isolation
DNA extraction
Genaidd Korea – I googled this and no kit came up. More details on kit name and company. If readers can’t find the kit, then they can not know what the method was.
What medical gauge? Or is this meant to be gauze? And then I would still like to know size exclusion
How was it washed? Saline added, centrifuged (or naturally sedimented?) and supernatant removed?
“Finally, formal-ether sedimentation techniques were followed” at minimum reference the technique used. There are several formal-ether techniques
PCR
Previously described – reference to where they were.
Company in brackets for reagents
What was the final concentration of primer? That is more useful than volume used.
Results
400 samples – back to my question from methods, were these all environmental samples? I
Table 1
Sulaimani center, Halabja, and Garmian district – so where are Kalar and Rzgari coming into this?
Statistical analysis in this table means nothing, what test? You need to put all stats used in the methods.
% numbers in table – put in a separate column and indicate that it is overall % positive, not for each area.
It really isn’t prevalence of stray dogs, it’s prevalence of stool samples – which I am assuming were all environmental and not tagged back to any specific dogs? Relative number of stray dogs in each area might be interesting to know.
“The results are in agreement with Mero et al. [20] who recorded the prevalence rate (11.5%) of Echinococcosis among slaughtered animals” None of the slaughtered animals were dogs, so not sure of relevance here. Particularly as you are not finding prevalence of dogs but prevalence of environmental stool samples. Instead of saying agree, state that the prev in stock animals in the same area is ---% and then link it to dogs becoming infected.
“The direct access of the infected organs of animals for stray dogs was suggested to be the main cause of the high prevalence rate among stray dogs. Similar results were reported by Kohansal et al. [21] in Iran (19.1%).” What similar results? Access to organs, prevalence in dogs?
Hydatid not hydrated
Sentence structure in this paragraph (first para page 5) needs works
“Many studies in different countries reported the highest rate of Echinococcosis among stray dogs than our finding.” Not a good sentence. Rather than saying in this study it was higher, in this it was lower, state prev varies from highest % to lowest % in country and country respectively. Were these other studies using the same method of looking at environmental samples? Dogs? Other animals? Make sure they really are comparable to this study.
Reference 8 doesn’t state that they identified specifically echinococcus, just taenia species.
Reference 22 is a review, “recorded the highest prevalence rate in Iran” what was that prevalence rate? And since this ref is a review what was the year/period that this result is for?
“The high prevalence rate of taenia species among stray dogs” Be more specific to echinococcus, unless you mean other taeniid species which might also be in dogs
The current study is the first survey using dog feces examination to determine the prevalence rate of Echinococcosis among stray dogs in Sulaimani province.” So prevalence was unknown in this province previously?
I think the methods need to be clearer on what was done. All eggs were morphologically identified. Then only 50 samples where eggs were found were subjected to PCR, not all samples where eggs were found?
“Due to the morphologically identical of the taeniid eggs” High morphological similarity of taeniid eggs
“Many molecular investigations back up the present research [26] in Iran, [27] in China; [28] in Italy, [29] in India; and [30].” And what? I would put the ref after each country that it is for. Ref 28 is Greece. 30 is Iraq and not a molecular study.
Author Response
Response to Reviewer 2 comments
Introduction
Comment: I would start with echinococcus and lead into the hosts and open defecation of dogs particularly as a risk factor for human infection with echinococcosis – and other helminths. What do you mean by semisenescent?
Response: It means the larvae becoming more advanced (Please see line 62)
Comment: Page two: these sentences on egg survivability should be rearranged to flow better and make more sense. For instance the first sentence looking at 7C and 21C, is this dry or humid? It then switches in the third sentence to talk about eggs surviving for weeks on months in humid environments – presumably with a temp above the 21C from the first sentence.
Response: The sentences were rearranged and the condition was humid, the paragraph was revised (Please see lines 64-65)
Comment: Page 2 paragraph 2, is there a reference for increase in food borne infections linked to fresh vegetables? Reference 7 is a single Russian paper, if you are stating around the world you need to have references that reflect and give that information.
Response: Regarding the increase of food-borne infection linked to vegetable yeas the reference was added [10] And regarding reference 7 and the order of it changed to 11 and another reference was added [12], (Please see lines 75 and 77).
Comment: Page 2, third paragraph, Introduce the idea of genotypes before stating that each genotype may infect many hosts.
Response: The paragraph about the strain and the host infection by each type was added (Please see lines 81-83)
Comment: The disease echinococcosis does not need to be in italics
Response: We agree with the reviewer, all echinococcosis were changed to non-italic and highlighted
Comment: “Because of the importance of stray dogs in CE transmission” – you haven’t indicated that dogs are important in transmission yet. This is where I would introduce echinococcus at the beginning then introduce dogs and how they are important in transmission.
Response: The new paragraph was written about how the dog transmits the egg of echinococcus granulosus, "The Dog is the final host of echinococcus granulosus, and infected dog daily will be shed a huge number of the eggs and it was the main source of soil, food and vegetable contamination with E. granulosus eggs." (Please see lines 88-90)
Comment: “Human is the dead-end of the disease since the human-to-human or dog-to-human transmission of the illness is impossible” So dogs are not important to transmission after all? I think you mean human to dog (and it is possible, just unlikely as it would require a dog to eat a human). Also the reference would not seem to support this claim.
Response: The sentences were rewritten and this phrase was added to be more clear "that due to it is unlikely as it would require a dog to eat a human" (Please see lines 94-95)
Comment: “Livestock CE is widespread in many regions of Asia including Middle East countries (Iran, Iraq, Pakistan, and Saudi Arabia), Central Asia (Kazakhstan, Kyrgyzstan, Tajikistan, Turkmenistan, and Uzbekistan), China, India, and Japan [13,14].” Refs 13 and 14 are for Nigeria and Iran. I assume they mention that it’s in other countries, but would be best to read some articles from these papers that are listed and then ref
Response: The paragraph was rewritten and some references were added, and the order of the references was changed (Please see line 98)
Comment: “In some cases, the finding suitable molecular genetic studies for strain identification of E. granulosus obtained primarily through the use of mitochondrial DNA (mtDNA),) which has an important role and is a new clue for the strategical control plan” reference 15 does not use molecular tools. So why is it referenced?
Response: It was a typo mistake. We have corrected and cited the relevant reference [21] (Please see line 105).
Comment: “The infected dog is the main source of environmental contamination with dog tapeworm egg” full stop here.
Response: The full stop was put (Please see line 106)
Comment: Reference 16 is not a specific reference to the ELISA. I also read the article and ELISA was not performed in that study, so why is this reference used?
Response: This is a mistake in reference numbering. The correct references were added [22] (Please see line 109).
Comment: This introduction is really confused and not at all focused. Consider the story you want to tell. Based on the title it should be an introduction to echinococcus in Iraq and how stray dogs may be important. The referencing throughout is quite poor.
Response: Many studies done in Iraq and Kurdistan region but according to our knowledge the study of stray dogs is inadequate in this region.
Methods
Comment: What year was the study performed
Response: The study was performed from September 2019 to November 2021. The year of the study was added (Please see line 125).
Comment: Collection of dog faeces. How many from each location?
Response: The samples were collected from 4 locations as follows: Sulaimani city (n= 130) samples, Halabja (n= 115) samples, Kalar (n= 120) samples and Rzgary (n=35) . The number per location was added to the section methodology. (Please see lines 124-125).
Comment: “The survey was not based on a certain number of dogs, but rather on the availability of Echinococcus granulosus eggs in the region”. Explain this further. So the prev of echinococcus was already known? If you are cherry picking locations based on having known high prev are you really finding the prevalence?
Response: The sentences were revised and the additional explanation was added as "The survey was not based on a certain number of dogs, but rather on the availability dog feces from different areas to find the rate and prevalence of E. granulosus among stray dogs" (Please see line 128-129).
Comment: How many sites?
Response: 20 sites from each area (the number of the site was added) (Please see line 129).
Comment: When you say dog feces samples, were these all environmental?
Response: The environment (soil) will be contaminated through dog feces
Comment: Once you start talking about recovering eggs you need to start at minimum a new paragraph, better, a new subheading for egg isolation
Response: The new subheading “Isolation of eggs” was added as per the suggestion of the reviewer (Please see line 134).
Comment: DNA extraction, Genaidd Korea – I googled this and no kit came up. More details on kit name and company. If readers can’t find the kit, then they can not know what the method was.
Response: The (Geneaid, Taiwan) is the correct name of the Company so this is a typo mistake corrected (Please see line 143). https://www.geneaid.com/about_intro.php
Comment: What medical gauge? Or is this meant to be gauze? And then I would still like to know size exclusion
Response: It was medical gauze (Size: 10.2-10.4 cm), corrected and the size was added (Please see lines 144-145).
Comment: How was it washed? Saline added, centrifuged (or naturally sedimented?) and supernatant removed?
Response: The feces washing is done by adding normal saline or PBS to the centrifuge tube 15 ml which contains about 200mg of feces then centrifuged for 5 minutes at 3000 rpm this process will be repeated three times or until the feces color becomes clear. The washing process is known so we think it does not need to mention all procedures. Additional explanation was added (Please see lines 146-147).
Comment: “Finally, formal-ether sedimentation techniques were followed” at minimum reference the technique used. There are several formal-ether techniques
Response:
PCR
Comment: Previously described – reference to where they were.
Response: The reference was added [26] (Please see line 147).
Comment: Company in brackets for reagents
Response: “Promega, USA” was added (Please see line 155)
Comment: What was the final concentration of primer? That is more useful than volume used.
Response: The final concentration of the primer was 10 pmol/uL. This information was added and highlighted (Please see line 156).
Results
Comment: 400 samples – back to my question from methods, were these all environmental samples? I
Response: Yes, all 400 samples of dog feces were collected from different environments (dog feces on the soil)
Comment: Table 1 Sulaimani center, Halabja, and Garmian district – so where are Kalar and Rzgari coming into this?
Response: The Garmian district is the common name used for Caller and Rzgari here in our culture your comment is right, we replaced Garmian with Kalar and Rzgari, corrected in the main text (Please see line 173).
Comment: Statistical analysis in this table means nothing, what test? You need to put all stats used in the methods.
Response: Yes, the statistical analysis revealed non-significant differences in prevalence or percentage of echinococcosis among stray dogs in the four local areas, it means there is no strategic planning for, controlling the stray dogs in our region and all sites of sampling have the same culture and service.
The test used is chi-square = χ2 = 5.58 (table 1).
Comment: % numbers in table – put in a separate column and indicate that it is overall % positive, not for each area.
Response: Done (Please see Table 1)
Comment: It really isn’t prevalence of stray dogs, it’s prevalence of stool samples – which I am assuming were all environmental and not tagged back to any specific dogs? Relative number of stray dogs in each area might be interesting to know.
Response: Your comment is scientifically valuable, but unfortunately, in our region, we have no systematic data and we have no official survey to know the number of stray dogs and also owner dogs.
Comment: “The results are in agreement with Mero et al. [20] who recorded the prevalence rate (11.5%) of Echinococcosis among slaughtered animals” None of the slaughtered animals were dogs, so not sure of relevance here. Particularly as you are not finding prevalence of dogs but prevalence of environmental stool samples. Instead of saying agree, state that the prev in stock animals in the same area is ---% and then link it to dogs becoming infected.
Response: We agree with the reviewer. The statement was changed (Please see lines 189-192)
Comment: “The direct access of the infected organs of animals for stray dogs was suggested to be the main cause of the high prevalence rate among stray dogs. Similar results were reported by Kohansal et al. [21] in Iran (19.1%).” What similar results? Access to organs, prevalence in dogs?
Response: Corrected, "Similar results of high prevalence rate (19.1%) were reported by Kohansal et al. [29]" (Please see line 198).
Comment: Hydatid not hydrated
Response: Corrected (Please see line 206).
Comment: Sentence structure in this paragraph (first para page 5) needs works
Response: The paragraph was rewritten and it is now more clear (line 224-226).
Comment: “Many studies in different countries reported the highest rate of Echinococcosis among stray dogs than our finding.” Not a good sentence. Rather than saying in this study it was higher, in this it was lower, state prev varies from highest % to lowest % in country and country respectively. Were these other studies using the same method of looking at environmental samples? Dogs? Other animals? Make sure they really are comparable to this study.
Response: The suggested sentences were revised (Please see lines 245-247).
Comment: Reference 8 doesn’t state that they identified specifically echinococcus, just taenia species.
Response: We agree with the reviewer. Revised accordingly [14] (Please see line 81).
Comment: Reference 22 is a review, “recorded the highest prevalence rate in Iran” what was that prevalence rate? And since this ref is a review what was the year/period that this result is for?
Response: The mentioned article is a review article, but the authors calculated the average infection rate among stray dogs they state, "the average infection among 11593 dogs was estimated to be 11.28%" They collected all registered reports from 1985 to 2019 in Iran. This information was added (line 198-204).
Comment: “The high prevalence rate of taenia species among stray dogs” Be more specific to echinococcus, unless you mean other taeniid species which might also be in dogs
Response: The taenia was changed to echinococcus species (line 186).
Comment: The current study is the first survey using dog feces examination to determine the prevalence rate of Echinococcosis among stray dogs in Sulaimani province.” So prevalence was unknown in this province previously?
Response: Yes, there are many limited studies In the Kalar district on Stray dogs and there are no adequate data about the prevalence of dog tapeworm in Sulaiamani province.
Comment: I think the methods need to be clearer on what was done. All eggs were morphologically identified. Then only 50 samples where eggs were found were subjected to PCR, not all samples where eggs were found?
Response: We agree with the reviewer. we confirmed 50 positive samples to investigate are these eggs belong to which type of taenia species and the PCR product were sequenced and the sequences submitted to Genebank during the article submission we not received the accession number so we exclude the sequence part now the accession number released and we added the accession numbers (OM200173, OM200174, OM200175, OM200176, OM200177) all sequences belong to the Echinococcus granulous, this result support our finding microscopically the most taenia eggs belong to the echinococcus spp (Please see lines 228-235).
Comment: “Due to the morphologically identical of the taeniid eggs” High morphological similarity of taeniid eggs
Response: Ok the high morphological similarity is more accurate morphological identical changed to High morphological similarity (Please see line 225).
Comment: “Many molecular investigations back up the present research [26] in Iran, [27] in China; [28] in Italy, [29] in India; and [30].” And what? I would put the ref after each country that it is for. Ref 28 is Greece. 30 is Iraq and not a molecular study.
Response: We agree with the reviewer. The references were changed and put after each country. Ref. 32 was done in Iraq and Italy changed to Greece (Please see lines 222-223).
Finally, we appreciate your scientific effort and your comments surely improved our article.

Reviewer 3 Report
The work concerns an important and interesting problem, but I have some comments regarding the content of the manuscript.
The title is inconsistent with the results of the work.
The applied PCR method without sequencing does not show the molecular characteristics but is only molecular diagnostics.
In the introduction, the author mentions genospecies (G1- to G10) but did not show which ones were found.
In conclusion, the phrase "the prevalence and the percentage of Echinococcus granulosus" is a synonym.
There are many inaccuracies in the introduction.
"A dog or a cat is often seen in close quarters with humans" -
the cat is not a host for E. granulosus.
"Various kinds of microorganisms potentially harmful to humans ..." - tapeworms are not microorganisms.
"On expulsion, the maturing process of the eggs of the dog tapeworm ...." Taenidae eggs are invasive already at the moment of expulsion from the host's organism.
"When these are semisenescent, these may immunize the intermediate host, but these cannot become larvae" only a live larva can hatch from the egg and immunize the host.
Dead larvae do not hatch and immunize.
"The growth and development of E. granulosus were slowed by the cold environmen" - Taenide eggs do not develop, only in a cold environment they survive longer.
There is no logical thread in the introduction. There is only cut-in information. I suggest that the introduction be thoroughly edited. Transfer some information to the discussion.
As for the methodology.
It is unclear whether the results are the mean of flotation and decantation. Were all samples analyzed by two methods.
Why is the genotype not known. In fact, the study lacks the molecular characteristics of E. granulosus.
Author Response
Response to Reviewer 3 comments
Comment: The work concerns an important and interesting problem, but I have some comments regarding the content of the manuscript.
Response: Thanks for your valuable comment
Comment: The title is inconsistent with the results of the work.
Response: We agree with the reviewer, but the molecular part, especially the sequencing of the DNA was excluded when we submitted the article because we did not get the access number from GeneBank. Fortunately, now we have the accession numbers of our sequences, and all accession numbers were added and discussed accordingly (Please see lines 33-35)
Comment: The applied PCR method without sequencing does not show the molecular characteristics but is only molecular diagnostics.
Response: The sequencing and accession numbers were analyzed and added (Please see lines 161-163 and lines 226-230)
Comment: In the introduction, the author mentions genospecies (G1- to G10) but did not show which ones were found.
Response: The paragraph about the genotyping and G1-G10 were added (Please see lines 81-83).
Comment: In conclusion, the phrase "the prevalence and the percentage of Echinococcus granulosus" is a synonym.
Response: We agree with the reviewer. “percentage” was removed (Please see line 18)
Comment: There are many inaccuracies in the introduction.
"A dog or a cat is often seen in close quarters with humans" - the cat is not a host for E. granulosus.
Response: We are with the reviewer that the cat cannot become the final host look like Dog, but some case reports indicate the cat can be infected with echinococcosis as an intermediate host, please see the following references
- Armua-Fernandez MT, Castro OF, Crampet A, Bartzabal Á, Hofmann-Lehmann R, Grimm F, Deplazes P. First case of peritoneal cystic echinococcosis in a domestic cat caused by Echinococcus granulosus sensu stricto (genotype 1) associated with feline immunodeficiency virus infection. Parasitology International. 2014 Apr 1;63(2):300-2.
- Oguz B, Selcin O, Deger MS, Bicek K, Ozdal N. A Case Report of Echinococcus granulosus sensu stricto (G1) in a Domestic Cat in Turkey. Journal of the Hellenic Veterinary Medical Society. 2021;72(4):3529-34.
3 Bonelli P, Masu G, Dei Giudici S, Pintus D, Peruzzu A, Piseddu T, Santucciu C, Cossu A, Demurtas N, Masala G. Cystic echinococcosis in a domestic cat (Felis catus) in Italy. Parasite. 2018;
Comment: "Various kinds of microorganisms potentially harmful to humans ..." - tapeworms are not microorganisms.
Response: We agree with the reviewer. The microorganism changed to pathogens which include both macro and microorganism (Please see line 47).
Comment: "On expulsion, the maturing process of the eggs of the dog tapeworm ...." Taenidae eggs are invasive already at the moment of expulsion from the host's organism.
Response: The mentioned sentence changed accordingly (Please see line 60).
Comment: "When these are semisenescent, these may immunize the intermediate host, but these cannot become larvae" only a live larva can hatch from the egg and immunize the host. Dead larvae do not hatch and immunize.
Response: Yes, We agree with your comment. The semi-senescent means the developed larvae not mature larvae, and it is alive. Changes accordingly
Comment: "The growth and development of E. granulosus were slowed by the cold environmen" - Taenide eggs do not develop, only in a cold environment they survive longer.
Response: The sentence is rewritten and the phrase "while it can survive for a long time in such a condition" was added (Please see line 69).
Comment: There is no logical thread in the introduction. There is only cut-in information. I suggest that the introduction be thoroughly edited. Transfer some information to the discussion.
Response: The introduction was revised according to the reviewer's comments.
Comment: As for the methodology. It is unclear whether the results are the mean of flotation and decantation. Were all samples analyzed by two methods.
Response: The methodology was revised yes, both methods were followed but flotation was sometimes used not for all samples the flotation was removed and only direct and concentration methods (formal ether ) were included because all samples were subjected to direct and sedimentation techniques (lines 135-136).
Comment: Why is the genotype not known. In fact, the study lacks the molecular characteristics of E. granulosus.
Response: We have added the sequence analysis and accession number of our sequences in the revised manuscript (Please see lines 221-233).
Thanks for your valuable comments

Round 2
Reviewer 2 Report
The English still needs to be improved particularly in the Introduction and Resutls/Discussion
Abstract
Line 23: Remove the i.e. and put a comma after techniques, and then after (formal-ether)
Line 32: delete ‘for the suspected band 44bp’
Line 34-35: I’d probably remove the accession numbers from here and just have them in the main body of the manuscript.
Introduction
Line 46-58: This is interesting info, but I would focus on dogs rather than cats – perhaps you can add some dog info in here to complement the cat info, or remove the cat info and add only dog info here. Personally I would briefly mention cats and focus on dogs as this is the focus of the paper.
Line 60: “On expulsion, the eggs of the dog tapeworm begin in the proglottides”. To me, the dog tapeworm is Diplydium caninum. Thus, instead of dog tapeworm change to Echinococcus. I would also start with proglottids.
“Upon expulsion proglottids, which contain eggs, of Echinococcus are exposed to the external environment.”
I would delete the next two sentences.
Lines 73-8: How is this paragraph contributing to the rest of the paper? If you have it in here, it needs to make sense for the paper. I would delete.
Line 94-95: This still says dog-to-human. If dog-to-human is impossible, then why are you doing the study? Change to human-to-dog transmission. Rather than illness say parasite.
Delete Due to this, it is unlikely, as it would require a dog to eat a human
I would move lines 79-98 to the beginning of the introduction, then have the cat paragraph (which should be changed to a dog paragraph)
Materials and methods
Geneaid, Taiwan – this time I did find the company so that is good. Put the actual name of the kit used as well. i.e. was it the stool extraction kit?
Line 161: sent to what company?
You still need to put a section for statistical analysis here where you can put what software you used to calculate all stats (prevalence, chi squared etc)
Still do not mention that only a subset of samples were amplified by PCR. This needs to go in the methods. Best place would be beginning of DNA extraction section. Something like, “In total 50 samples, where eggs were isolated as described above, had DNA extracted”.
Results and Discussion
Table 1: ‘statistical analysis’ is meaningless, particularly as no where do you say what analysis you did. Put Chi squared as the heading. Put the stats in the methods – what program you used to calculate etc.
Line 190: may be due to a lack of proper management – you don’t know 100% this is the case.
Line 208: ‘and sold illegally’
Line 212: “may be due to stray dogs which are running free and living in the street, and can easily reach the offal etc etc etc”
Line 222: Really need to have this information in the methods
Table 2: Positive by PCR as heading, instead of PCR examination
Line 237-238: “is valuable in characterizing molecular genotypes of Echino species.” I would have a full stop then and have the next bit as a separate sentence.
Line 241: Delete ‘with an infection rate of 80%’. It’s not an infection rate. An infection rate can also be called an incident rate, and it’s the risk of infection in a population. You could but % in brackets after 40, but also indicate this is of 50 samples total.
Conclusions
Line 252: ‘most important species’ – otherwise, what others
Author Response
Response to Reviewer comments
Comment: The English still needs to be improved particularly in the Introduction and Resutls/Discussion
Response: We gave crosschecked the whole manuscript regarding any spelling and grammar mistakes and highlighted as yellow were revised/corrected.
Abstract
Comment: Line 23: Remove the i.e. and put a comma after techniques, and then after (formal-ether)
Response: The i.e. removed and a comma was put after techniques and after (formal-ether),
Comment: Line 32: delete ‘for the suspected band 44bp’
Response: deleted
Comment: Line 34-35: I’d probably remove the accession numbers from here and just have them in the main body of the manuscript.
Response: The accession numbers were removed in the abstract.
Introduction
Comment: Line 46-58: This is interesting info, but I would focus on dogs rather than cats – perhaps you can add some dog info in here to complement the cat info, or remove the cat info and add only dog info here. Personally I would briefly mention cats and focus on dogs as this is the focus of the paper.
Response: Revised as per the recommendation of the reviewer
Comment: Line 60: “On expulsion, the eggs of the dog tapeworm begin in the proglottides”. To me, the dog tapeworm is Diplydium caninum. Thus, instead of dog tapeworm change to Echinococcus. I would also start with proglottids.
Response: Revised
Comment: “Upon expulsion proglottids, which contain eggs, of Echinococcus are exposed to the external environment.”
I would delete the next two sentences.
Response: Deleted
Comment: Lines 73-8: How is this paragraph contributing to the rest of the paper? If you have it in here, it needs to make sense for the paper. I would delete.
Response: Deleted
Comment: Line 94-95: This still says dog-to-human. If dog-to-human is impossible, then why are you doing the study? Change to human-to-dog transmission. Rather than illness say parasite.
Response: Revised accordingly
Comment: Delete Due to this, it is unlikely, as it would require a dog to eat a human
I would move lines 79-98 to the beginning of the introduction, then have the cat paragraph (which should be changed to a dog paragraph)
Response: The " Due to this, it is unlikely, as it would require a dog to eat a human" was deleted and the paragraph from line 79-98 were moved to the beginning of the introduction.
Materials and methods
Comment: Geneaid, Taiwan – this time I did find the company so that is good. Put the actual name of the kit used as well. i.e. was it the stool extraction kit?
Response: The name of the kit is The stool DNA extraction kit added
Comment: Line 161: sent to what company?
Response: Macrogen company Korea, added
Comment: You still need to put a section for statistical analysis here where you can put what software you used to calculate all stats (prevalence, chi squared etc)
Response: The section 2.6 statistical analysis was added
Comment: Still do not mention that only a subset of samples were amplified by PCR. This needs to go in the methods. Best place would be beginning of DNA extraction section. Something like, “In total 50 samples, where eggs were isolated as described above, had DNA extracted”.
Response: The sentences revised
Results and Discussion
Comment: Table 1: ‘statistical analysis’ is meaningless, particularly as no where do you say what analysis you did. Put Chi squared as the heading. Put the stats in the methods – what program you used to calculate etc.
Response: The chi-square test at the heading.
Comment: Line 190: may be due to a lack of proper management – you don’t know 100% this is the case.
Response: Revised.
Comment: Line 208: ‘and sold illegally’
Response: added sold illegally
Comment: Line 212: “may be due to stray dogs which are running free and living in the street, and can easily reach the offal etc etc etc”
Response: The sentences changed to (maybe due to stray dogs which are running free and living in the street, and can easily reach the offal and infected organs of the slaughtered animals).
Comment: Line 222: Really need to have this information in the methods
Response: The sentences revised
Comment: Table 2: Positive by PCR as heading, instead of PCR examination
Response: yeas the PCR Examination changed to Positive by PCR.
Comment: Line 237-238: “is valuable in characterizing molecular genotypes of Echino species.” I would have a full stop then and have the next bit as a separate sentence.
Response: Full stop was added.
Comment: Line 241: Delete ‘with an infection rate of 80%’. It’s not an infection rate. An infection rate can also be called an incident rate, and it’s the risk of infection in a population. You could but % in brackets after 40, but also indicate this is of 50 samples total.
Response: The sentences were deleted and the paragraph revised
Conclusions
Comment: Line 252: ‘most important species’ – otherwise, what others
Response: The common species is E. Granulosis but we mean the important strains that are responsible for human echinococcosis are sensu lato (G1-G3), the sentences removed.
Finally, we would like to thank you for your valuable and scientific comment and recommendation.

Round 3
Reviewer 2 Report
Line 33-34: delete ‘the DNA sequences deposited in GeneBank’
Line 74-76: I would delete this because it doesn’t make sense “An oncosphere must develop enough to become a larva [3]. When 74 these are semi-senescent, these may immunize the intermediate host, but these cannot be-75 come larvae”
Line 78: delete on the other hand
Author Response
Response to the reviewer comments
Line 33-34: delete ‘the DNA sequences deposited in GeneBank’
Response: Deleted
Line 74-76: I would delete this because it doesn’t make sense “An oncosphere must develop enough to become a larva [3]. When 74 these are semi-senescent, these may immunize the intermediate host, but these cannot be-75 come larvae”
Response: Deleted
Line 78: delete on the other hand
Response: Deleted
